organic chemistry

thermochromism, spiro-heterocycles, spiropyran, colorimetric

**Author for correspondence:**
Anila Iqbal
e-mail: a.anilaiqbal@gmail.com

This article has been edited by the Royal Society of Chemistry, including the commissioning, peer review process and editorial aspects up to the point of acceptance.

# Synthesis of novel silica encapsulated spiropyran-based thermochromic materials

Anila Iqbal[1], Ghazala Iqbal[2], Muhammad Naveed Umar[3], Haroon ur Rashid[4] and Sher Wali Khan[5]

[1]Nanoscience and Technology Department, National Center for Physics, PO Box No 2141, Islamabad 44000, Pakistan
[2]Department of Pharmacy, Kohat University of Science and Technology, (KUST), Kohat, KP, Pakistan
[3]Department of Chemistry, University of Malakand, Chakdara Dir (L), KP, Pakistan
[4]Institute of Chemistry, Federal University of Mato Grosso do Sul, Campo Grande, MS, Brazil
[5]Department of Chemistry, Shaheed Benazir Bhutto University, Sheringal, Upper Dir, KP, Pakistan

A series of novel spiropyrans were synthesized through the condensation of substituted 3,3-dimethyl-2-methyleneindoline with different nitro-substituted *o*-hydroxy aromatic aldehydes. Indoles were initially substituted with a variety of alkanes and esters moieties. The substituted 3,3-dimethyl-2-methyleneindoline was then reacted with nitro-substituted *o*-hydroxy aromatic aldehydes to yield the respective spiropyrans. The synthesized novel spiropyrans were encapsulated in silica nano-shells to protect them from the effect of moisture and pH. The thermochromic behaviour of novel spiropyrans was studied by UV-visible spectroscopy. The thermally induced isomerization of spiropyran derivatives was carried out in a water/ethanol mixture. The thermal isomerization of spiro-heterocyclic (colourless form) to merocyanine (MC) (coloured form) was a discontinuous process and was observed in a temperature range of 5–60°C via UV-visible spectrometer. The absorption process occurs reversibly regardless of the heating/cooling sequence. The spiropyran derivatives, therefore, have a potential application for colorimetric temperature indication.

## 1. Introduction

The phenomenon of thermochromism has been known since 1926 when the reversible colour change was observed with temperature [1]. Therefore, thermochromism can be defined as the reversible thermo-transformation of chemical species between two forms

**Figure 1.** Transformation of spiropyran.

giving rise to change in colour [2–4]. Thermochromism is the property of many organic, inorganic, organometallic, macromolecules [5–7] polymers [8,9] or gels [10]. Some of the supermolecular systems like liquid crystals also show thermochromism. Thermochromic compounds have been used in various applications such as imaging systems, temperature indicative devices, optical switching, etc. Among the organic molecules, spiro-heterocycles, Schiff bases, Bianthrone and overcrowded ethylenes are important thermochromic materials [11–14]. Spiropyrans are one of the contenders of the most versatile compound reported as they show diverse properties such as photochromism, fluorescence, mechanochromism, acidochromism, etc. [15]. All these properties have wide range of applications but photochromism and thermochromism of spiropyrans have attracted wide range of applications such as in cosmetic, fabrics, etc. [16]. Spiropyrans are considered as a good candidate for application in drug delivery systems in which they are used for incorporation with different polymeric micelles, polymersomes and polymeric nanoparticles (NPs), etc. [17]. Spiropyrans represent an important class of spiro-heterocycles in which an equilibrium exists between the colourless spiro-heterocyclic (SP) form and the coloured open merocyanine (MC) form through the cleavage of the C–O bond [15,18] as shown in figure 1. Various compounds such as silica, ionic liquids and fluoro alcohols have been used to stabilize MC form which is moisture- and pH-sensitive [15,19].

The main aim of encapsulation is to cloak any material (solid, liquid or gas) within the coating material. The purposes of encapsulation can be different and depend on the specific applications [20]. The main objective is always to protect the active material from external environment, which can affect the properties and applications of the main product (temperature, pH, etc.). It also helps to enhance the durability of the product by preventing the oxidation. Because of encapsulation, the core material properties are elevated [21]. Different encapsulation techniques result in the formation of different capsules and are classified by their size, i.e. microcapsules refer to the encapsulation where the size of these shells is less than 1 µm to 1000 µm in diameter and is of any type such as tubes and spheres, and the nanocapsules are synthesized at the nanoscale level [22]. Microencapsulation techniques consist of two main steps: emulsion formation/active ingredient suspension and shell formation. Emulsion preparation is a critical step as it determines the size distribution of the capsules, which are influenced by the operating conditions (the rate and time of agitation, viscosity, the mass ratio of different phases, etc.). Concentration of shell materials, pH, temperature and solubility effects the formation of the shell [23]. Different encapsulations are used in pharmaceutical [24], cosmetic [25] and food industries [26]. Microencapsulation methods are used for encapsulating different thermochromic leuco dyes, and further incorporation of these microcapsules in smart coatings has been developed, which aids in making sustainable buildings that use minimum power for heating and cooling applications [27].

The microencapsulation of thermochromic materials by amorphous silica could protect it from UV light (290–320 nm) [28]. A great selection of NPs such as $Fe_3O_4$, $SiO_2$, $ZrO_2$, $Al_2O_3$ and $TiO_2$ [29–31] have been used for coating. Leuco dyes have been encapsulated within silica to be used for energy saving in proposed building applications [32], poly(methylmethacrylate)/thermochromic system (PMMA/TS) microcapsules are synthesized by microencapsulation method and the thermochromic system was consisting of crystal violet lactone as a leuco dye, bisphenol A as a colour developer, and 1-tetradecanol as a solvent. In the case of PMMA/TS systems, the ratio of PMMA to TS plays a critical role, as the amount of PMMA increases, the change in colour become less visible due to the thickness of the shells. In comparison with the spiropyran, these TS are multi-component complex systems, which melt upon heating to 45–50°C [33]. In the case of spiropyrans, no such encapsulation within the silica shells is reported, whereas research has been carried out on the encapsulation of a spiropyrans within a self-assembled cage [34].

The use of silica spheres for encapsulation has received considerable attention due to their unique properties such as large surface area, high loading capacity, low density, in addition to their applications as microreactors, in drug storage, waste removal, chromatography, etc. [35]. In this study, 12 different types of spiropyrans were synthesized. In the first step, various N-substituted indoles were synthesized by using 2,3,3-trimethylindolene and alkyl halides (such as ethyl bromide, n-propyl

bromide, n-butyl bromide, iso-butyl bromide) or α-halo esters (such as methyl bromoacetate and ethyl bromoacetate). The substituted indoles were then reacted with 4-nitro-2-hydroxy benzaldehyde and 5-nitro-2-hydroxy benzaldehyde to get the respective spiropyrans. The fully characterized novel spiropyrans dyes were encapsulated in the silica shells through polymerization of tetraethyl orthosilicate (TEOS) in a basic media by using modified Stober method without using a surfactant. The Stober method involves hydrolysis and condensation polymerization with TEOS under basic conditions as shown in the following equations:

$$\equiv Si - OR + H_2O \rightleftarrows \; \equiv Si - OH + ROH, \quad \equiv Si - OR + H_2O \rightleftarrows \; \equiv Si - OH + ROH, \tag{1.1}$$

$$\equiv Si - OR + \; \equiv Si - OH \rightleftarrows \; \equiv Si - O - Si + ROH, \quad \equiv Si - OR + \; \equiv Si - OH \rightleftarrows \; \equiv Si - O - Si + ROH, \tag{1.2}$$

$$\equiv Si - OH + \; \equiv Si - OH \rightleftarrows \; \equiv Si - O - Si + H_2O, \quad \equiv Si - OH + \; \equiv Si - OH \rightleftarrows \; \equiv Si - O - Si + H_2O, \tag{1.3}$$

where R is an alkyl group $C_2H_5$. In the hydrolysis reaction (equation (1.1)), TEOS hydrolyses to generate siloxane molecules and ethanol. Then, the condensation polymerization occurs through the siloxane molecules and TEOS molecules (equation (1.2)) or siloxane molecules themselves (equation (1.3)).

We also planned the thermochromic behaviour of spiropyran dyes in an aqueous ethanol solution. The silica encapsulation was proposed for the ease of practical application of material over a wide range of surfaces without the effect of moisture and pH on the structure of the dye and henceforth the thermochromic behaviour. The prepared silica-encapsulated spiropyrans have a wide range of applications as temperature monitoring sensors which are used in motors, circuit breakers, heat exchangers and transformers. These can be applied in health indicators and to report food quality. The general applications involve decorative use on cloths, utensils, paper, etc. [36].

# 2. Experimental

## 2.1. Materials

All the reagents were purchased from Sigma Aldrich. The compounds were characterized by their physical constants data and spectro-analytical techniques. Melting points of the synthesized compounds were recorded in open capillaries using Gallenkamp melting point apparatus (MP-D). The chemical structure of synthesized compounds was characterized by infrared (IR), [1]H nuclear magnetic resonance ([1]H NMR) and [13]C nuclear magnetic resonance ([13]C NMR) spectroscopic techniques. The IR spectra were recorded on Thermoscientific Nicolet model 6700. [1]H NMR and [13]C NMR spectra were recorded on Bruker AV-300 MHz spectrometer using the desired solvent as indirect reference. UV-Vis spectroscopy was carried out at different temperatures using Shimadzu spectrophotometer, model Pharmaspec UV-1700. Surface morphology was studied by scanning electron microscopy (SEM) (JEOI, JSM-5910), operating at 20 kV with different magnification power.

## 2.2. Synthesis

The general procedure for the synthesis of intermediates and spiropyran derivatives is shown in figure 2.

## 2.3. General procedure for synthesis of substituted indole

First, 0.795 g (5 mmol) of 2,3,3-trimethyl-3*H*-indole was dissolved in dry distilled acetonitrile (15 ml), and then 2.18 g (20 mmol) of alkyl halide was added to the reaction mixture. The reaction was allowed to reflux for 78 h, and the progress was monitored by thin-layer chromatography (TLC) analysis till the completion of the reaction. The reaction mixture was then cooled to room temperature and concentrated under reduced pressure. The residue was added to n-hexane (15 ml) and dispersed well by ultra-sonication. The insoluble solid was recovered by filtration. Solid KOH (4.0 mmol) was added to water and stirred at room temperature for 10 min. The solution was extracted with diethyl ether, washed with distilled water, brine, dried over anhydrous $Na_2SO_4$, and the excess solvent was removed under reduced pressure.

### 2.3.1. Synthesis of 1-ethyl-3,3-dimethyl-2-methyleneindoline (TH-1)

The compound (TH-1) was synthesized by the same general procedure as given above, using 2,3,3-trimethyl-3*H*-indole (0.795 g, 5 mmol), dry distilled acetonitrile (15 ml), ethyl bromide (2.18 g, 20 mmol) and KOH (4.0 mmol).

**Figure 2.** Synthetic scheme for the synthesis of spiropyrans and their encapsulation.

The light orange liquid was obtained with a yield of 80%. TH-1 was used without further purification in the next step.

### 2.3.2. Synthesis of 3,3-dimethyl-2-methylene-1-propylindoline (TH-2)

The compound (TH-2) was synthesized by the same general procedure as given before, using 2,3,3-trimethyl-3H-indole (0.795 g, 5 mmol), dry distilled acetonitrile (15 ml), propyl bromide (1.623 g, 10 mmol) and KOH (4.0 mmol).

Dark pink oily product was obtained with a yield of 79%. This product was used in the next step without further purification.

### 2.3.3. Synthesis of 1-butyl-3,3-dimethyl-2-methyleneindoline (TH-3)

The compound (TH-3) was synthesized by the same general procedure as given before, using 2,3,3-trimethyl-3H-indole (0.795 g, 5 mmol), dry distilled acetonitrile (15 ml), n-butyl bromide (2.055 g, 15 mmol) and KOH (4.0 mmol).

Yellow oily product was obtained with a yield of 86%. This product was used in the next step without further purification.

### 2.3.4. Synthesis of 1-sec-butyl-3,3-dimethyl-2-methyleneindoline (TH-4)

The compound (TH-4) was synthesized by the same general procedure as given before, using 2,3,3-trimethyl-3H-indole (0.795 g, 5 mmol), dry distilled acetonitrile (15 ml), iso-butyl bromide (3.95 g, 28.8 mmol) and KOH (4.0 mmol).

Pink oily product was obtained with a yield of 95%. This product was used in the next step without further purification.

### 2.3.5. Synthesis of methyl-2-(3,3-dimethyl-2-methyleneindolin-1-yl)acetate (TH-5)

The compound (TH-5) was synthesized by the same general procedure as given before, using 2,3,3-trimethyl-3H-indole (0.159 g, 1 mmol), dry distilled acetonitrile (15 ml), methyl bromoacetate (0.832 g, 2.16 mmol) and KOH (4.0 mmol).

Blackish brown oily product was obtained with a yield of 72%. This product was used in the next step without further purification.

## 2.3.6. Synthesis of methyl-3-(3,3-dimethyl-2-methyleneindolin-1-yl)propanoate (TH-6)

The compound (TH-6) was synthesized by the same general procedure as given before, using 2,3,3-trimethyl-3H-indole (0.795 g, 5 mmol), dry distilled acetonitrile (15 ml), ethyl bromo acetate (1.135 g, 6.79 mmol) and KOH (4.0 mmol).

Brown oily product was obtained with a yield of 91%. This product was used in the next step without further purification.

## 2.4. General procedure for synthesis of spiropyrans

Substituted indole was dissolved in 15 ml of dry, distilled ethanol under a static nitrogen atmosphere. The mixture was set to gentle reflux with stirring followed by the addition of substituted benzaldehyde. The reaction was allowed to stir under reflux for 12 h, and the progress was monitored by TLC until the reaction was complete. The product was recovered by filtration, washed with ethanol and dried in a vacuum. International Union of Pure and Applied Chemistry (IUPAC) names and chemical structures of the synthesized compounds are given in table 1.

### 2.4.1. Synthesis of 1′,3′-dihydro-3′,3′-dimethyl-1′-ethyl-7-nitro-spiro(2H-1-benzopyran-2,2′-(2H)-indole) (SP-1)

The compound (SP-1) was synthesized by the same general procedure as given above, using 1-ethyl-3,3-dimethyl-2-methyleneindoline (TH-1) (0.325 g, 1.73 mmol), EtOH (15 ml) and 2-hydroxy-4-nitrobenzaldehyde (0.300 g, 1.79 mmol).

Dark brown solid was obtained with a yield of 90%. **IR** ($\bar{v}_{max}$, cm$^{-1}$) 1650 (C=C aliphatic), 1609 (C=C aromatic), 1332 (C–N), 1213 (C–O cyclic ether). **$^1$H NMR** (300 MHz, DMSO): $\delta$ (ppm) = 1.08 (t, 3H, 7.5 Hz, –NCH$_2$CH$_3$), 1.08 (s, 3H, –CCH$_3$), 1.20 (s, 3H, –CCH$_3$), 3.16 (q, 2H, 7.2 Hz, –NCH$_2$CH$_3$), 6.00 (d, 1H, 10.5 Hz, –CCHCH), 6.60 (d, 1H, 8.5 Hz, ArH), 6.77 (t, 1H, 8.7 Hz, ArH), 6.86 (d, 1H, 9 Hz, ArH), 7.11 (d, 1H, 8.4 Hz, ArH), 7.11 (t, 1H,8.9 Hz, ArH), 7.20 (d, 1H, 10.5 Hz, –CHCHC), 7.98, 8.00 (dd, 1H, 2.7, 9 Hz, ArH), 8.22 (d, 1H, 2.7 Hz, ArH). **$^{13}$C NMR** (75 MHz, DMSO): $\delta$ (ppm) = 14.0, 19.9, 22.1, 45.0, 52.7, 106.8, 106.9, 115.8, 119.1, 119.3, 122.1, 122.3, 123.3, 126.1, 128.0, 128.5, 136.0, 140.8, 147.2, 159.7.

### 2.4.2. Synthesis of 1′,3′-dihydro-3′,3′-dimethyl-1′-propyl-7-nitro-spiro(2H-1-benzopyran-2,2′-(2H)-indole) (SP-2)

The compound (SP-2) was synthesized by the same general procedure as given before, using 3,3-dimethyl-2-methylene-1-propylindoline (TH-2) (0.404 g, 2.0 mmol), EtOH (15 ml) and 2-hydroxy-4-nitrobenzaldehyde (0.207 g, 1.23 mmol).

Ecru solid was obtained with a yield of 92%. **IR** ($\bar{v}_{max}$, cm$^{-1}$) 1648 (C=C aliphatic), 1608 (C=C aromatic), 1333 (C–N), 1209 (C–O cyclic ether). **$^1$H NMR** (300 MHz, DMSO): $\delta$ (ppm) = 0.84 (t, 3H, 7.2 Hz, –CH$_2$CH$_3$), 1.10 (s, 3H, –CCH$_3$), 1.19 (s, 3H, –CCH$_3$), 1.60, (m, 2H, 7.5 Hz, –CH$_2$CH$_2$CH$_3$), 3.08 (t, 2H, 7.2 Hz, –NCH$_2$CH$_2$), 6.00 (d, 1H, 10.5 Hz, –CCHCH), 6.59 (d, 1H, 7.5 Hz, ArH), 6.77 (t, 1H,7.5 Hz, ArH), 6.86 (d, 1H, 9 Hz, ArH), 7.11 (t, 1H, 8.9 Hz, ArH), 7.06, 7.11 (dd, 1H, 6.6, 8.4 Hz, ArH), 7.20 (d, 1H, 10.5 Hz, –CHCHC), 7.97,7.99 (dd, 1H, 2.7, 9 Hz, ArH), 8.21 (d, 1H, 2.7 Hz, ArH). **$^{13}$C NMR** (75 MHz, DMSO, TMS): $\delta$ (ppm) = 11.9, 19.9, 22.1, 26.3, 45.0, 52.7, 106.8, 106.9, 115.8, 119.1, 119.3, 122.1, 122.3, 123.3, 126.19, 128.0, 128.5, 136.0, 140.8, 147.2, 159.7.

### 2.4.3. Synthesis of 1′,3′-dihydro-3′,3′-dimethyl-1′-butyl-7-nitro-spiro(2H-1-benzopyran-2,2′-(2H)-indole) (SP-3)

The compound (SP-3) was synthesized by the same general procedure as given before, using 1-butyl-3,3-dimethyl-2-methyleneindoline (TH-3) (0.663 g, 3.0 mmol), EtOH (15 ml) and 2-hydroxy-4-nitrobenzaldehyde (0.300 g, 1.79 mmol).

Citrine solid was obtained with a yield of 83%. **IR** ($\bar{v}_{max}$, cm$^{-1}$) 1652 (C = C aliphatic), 1608 (C=C aromatic), 1325 (C–N), 1213 (C–O cyclic ether). **$^1$H NMR** (300 MHz, DMSO): $\delta$ (ppm) = 0.83 (t, 3H, 7.5 Hz, –CH$_2$CH$_3$), 1.20 (s, 3H, –CCH$_3$), 1.24 (s, 3H, –CCH$_3$), 1.28 (sextet, 2H, 7.5 Hz, –CH$_2$CH$_2$CH$_3$), 1.52 (quintet, 2H, 7.5 Hz, –CH$_2$CH$_2$CH$_2$), 3.21 (t, 2H, 7.5 Hz, –NCH$_2$CH$_2$), 6.00 (d, 1H, 10.5 Hz, –CCHCH), 6.59 (d, 1H, 9 Hz, ArH), 6.77 (t, 1H, 9 Hz, ArH), 6.86 (d, 1H, 9 Hz, ArH), 7.10 (t, 1H,

**Table 1.** IUPAC names and chemical structures of the synthesized compounds.

| serial number | sample code | name | structure |
|---|---|---|---|
| 1 | SP-1 | 1′,3′-dihydro-3′,3′-dimethyl-1′-ethyl-7-nitro-spiro(2H-1-benzopyran-2,2′-(2H)-indole) | |
| 2 | SP-2 | 1′,3′-dihydro-3′,3′-dimethyl-1′-propyl-7-nitro-spiro(2H-1-benzopyran-2,2′-(2H)-indole) | |
| 3 | SP-3 | 1′,3′-dihydro-3′,3′-dimethyl-1′-butyl-7-nitro-spiro(2H-1-benzopyran-2,2′-(2H)-indole) | |
| 4 | SP-4 | 1′,3′-dihydro-1′-(1-methylpropyl)-3′,3′-dimethyl-7-nitro-spiro(2H-1-benzopyran-2,2′-(2H)-indole) | |
| 5 | SP-5 | 1′,3′-dihydro-1′-(1-methylacetate)-3′,3′-dimethyl-7-nitro-spiro(2H-1-benzopyran-2,2′-(2H)-indole) | |

(*Continued.*)

| serial number | sample code | name | structure |
| --- | --- | --- | --- |
| 6 | SP-6 | 1′,3′-dihydro-1′-(1-ethylacetate)-3′,3′-dimethyl-7-nitro-spiro(2*H*-1-benzopyran-2,2′-(2*H* )-indole) | |
| 7 | SP-7 | 1′,3′-dihydro-3′,3′-dimethyl-1′-ethyl-6-nitro-spiro(2*H*-1-benzopyran-2,2′-(2*H* )-indole) | |
| 8 | SP-8 | 1′,3′-dihydro-3′,3′-dimethyl-1′-propyl-6-nitro-spiro(2*H*-1-benzopyran-2,2′-(2*H* )-indole) | |
| 9 | SP-9 | 1′,3′-dihydro-3′,3′-dimethyl-1′-butyl-6-nitro-spiro(2*H*-1-benzopyran-2,2′-(2*H* )-indole) | |
| 10 | SP-10 | 1′,3′-dihydro-1′-(1-methylpropyl)-3′,3′-dimethyl-6-nitro-spiro(2*H*-1-benzopyran-2,2′-(2*H* )-indole) | |

| serial number | sample code | name | structure |
|---|---|---|---|
| 11 | SP-11 | 1′,3′-dihydro-1′-(1-methylacetate)-3′,3′-dimethyl-6-nitro-spiro(2H-1-benzopyran-2,2′-(2H)-indole) | |
| 12 | SP-12 | 1′,3′-dihydro-1′-(1-ethylacetate)-3′,3′-dimethyl-6-nitro-spiro(2H-1-benzopyran-2,2′-(2H)-indole) | |

9 Hz, Ar*H*), 7.12 (d, 1H, 9 Hz, Ar*H*), 7.21 (d, 1H, 10.5 Hz, –CH*C*HC), 7.98, 8.00 (dd, 1H, 2.7, 9 Hz, Ar*H*), 8.22 (d, 1H, 2.7 Hz, Ar*H*). $^{13}$C **NMR** (75 MHz, DMSO): $\delta$ (ppm) = 14.0, 18.0, 19.0, 19.9, 29.4, 48.9, 51.9, 106.0, 106.8, 107.0, 115.8, 119.9, 122.12, 122.1, 126.1, 127.9, 128.0, 128.5, 136.0, 141.0, 146.2, 158.0.

## 2.4.4. Synthesis of 1′,3′-dihydro-1′-(1-methylpropyl)-3′,3′-dimethyl-7-nitro-spiro(2H-1-benzopyran-2,2′-(2H)-indole) (SP-4)

The compound (SP-4) was synthesized by the same general procedure as given before, using 1-sec-butyl-3,3-dimethyl-2-methyleneindoline (TH-4) (0.54 g, 2.5 mmol), EtOH (15 ml) and 2-hydroxy-4-nitrobenzaldehyde (0.300 g, 1.79 mmol).

Dark golden rod solid was obtained with a yield of 80%. **IR** ($\bar{\upsilon}_{max}$, cm$^{-1}$) 1649 (C = C aliphatic), 1611 (C=C aromatic), 1335 (C–N), 1210 (C–O cyclic ether). $^{1}$**H NMR** (300 MHz, DMSO): $\delta$ (ppm) = 0.85 (t, 3H, 7.5 Hz, –CH$_2$C*H*$_3$), 1.09 (s, 3H, –CC*H*$_3$), 1.135 (s, 3H, –CC*H*$_3$), 1.25 (d, 3H, 7.4 Hz, –CHC*H*$_3$), 1.53 (quintet, 2H, 7.5 Hz, –CHC*H*$_2$CH$_3$), 2.79 (sextet, 1H, 7.8 Hz, –NC*H*(CH$_2$)(CH$_3$)), 6.03 (d, 1H, 10.5 Hz, –CC*H*CH), 6.59 (d, 1H, 9 Hz, Ar*H*), 6.83 (t, 1H, 8.9 Hz, Ar*H*), 6.89 (d, 1H, 9 Hz, Ar*H*), 7.10 (t, 1H, 9 Hz, Ar*H*), 7.25 (d, 1H, 10.5 Hz, –CH*C*HC), 8.03 (d, 1H, 3 Hz, Ar*H*), 8.11, 8.13 (dd, 1H, 3, 9 Hz, Ar*H*), 8.21 (d, 1H, 3 Hz, Ar*H*). $^{13}$C **NMR** (75 MHz, DMSO): $\delta$ (ppm) = 14.8, 19.9, 21.0, 22.1, 29.4, 49.8, 52.7, 106.0, 107.0, 115.8, 119.1, 119.4, 122.1, 122.1, 123.3, 127.0, 128.0, 128.5, 136.0, 141.0, 147.5, 159.6.

## 2.4.5. Synthesis of 1′,3′-dihydro-1′-(1-methylacetate)-3′,3′-dimethyl-7-nitro-spiro(2H-1-benzopyran-2,2′-(2H)-indole) (SP-5)

The compound (SP-5) was synthesized by the same general procedure as given before, using methyl-2-(3,3-dimethyl-2-methyleneindolin-1-yl)acetate (TH-5) (0.332 g, 1.43 mmol), EtOH (15 ml) and 2-hydroxy-4-nitrobenzaldehyde (0.207 g, 1.23 mmol).

Bright maroon solid was obtained with a yield of 80%. **IR** ($\bar{\upsilon}_{max}$, cm$^{-1}$) 1656 (C=C aliphatic), 1740 (C= O ether), 1607 (C=C aromatic), 1332 (C–N), 1212 (C–O cyclic ether), 119 (C–O ester). $^{1}$**H NMR** (300 MHz,

DMSO): $\delta$ (ppm) = 1.08 (t, 3H, 7.5 Hz, –OCH$_3$), 1.19 (s, 3H, –CCH$_3$), 1.30 (s, 3H, –CCH$_3$), 2.75 (s, 2H, –NCH$_2$C=O), 5.99 (d, 1H, 10.5 Hz, –CCHCH), 6.58 (d, 1H, 10.5 Hz, –CHCHC), 6.60 (d, 1H, 8.5 Hz, ArH), 6.817 (d, 1H, 9 Hz, ArH), 6.84 (t,1H, 8.7 Hz, ArH), 7.09 (t, 1H, 8.9 Hz, ArH), 7.15 (d, 1H, 8.6 Hz, ArH), 8.03, 8.05 (dd, 1H, 2.7,9 Hz, ArH), 8.12 (d, 1H, 2.7 Hz, ArH). $^{13}$C NMR (75 MHz, DMSO): $\delta$ (ppm) = 19.9, 22.1, 52.0, 52.7, 54.0, 106.8, 106.9, 115.8, 119.1, 119.3, 122.1, 122.3, 123.3, 126.1, 128.0, 128.5, 136.0, 140.8, 147.2, 159.7, 172.0.

### 2.4.6. Synthesis of 1′,3′-dihydro-1′-(1-ethylacetate)-3′,3′-dimethyl-7-nitro-spiro(2H-1-benzopyran-2,2′-(2H)-indole) (SP-6)

The compound (SP-6) was synthesized by the same general procedure as given before, using methyl-3-(3,3-dimethyl-2-methyleneindolin-1-yl)propanoate (TH-6) (0.812 g, 3.3 mmol), EtOH (15 ml) and 2-hydroxy-4-nitrobenzaldehyde (0.300 g, 1.79 mmol).

Light brown solid was obtained with a yield of 79%. **IR** ($\bar{\upsilon}_{max}$, cm$^{-1}$) 1639 (C=C aliphatic), 1746 (C=O ester), 1605 (C=C aromatic), 1336 (C–N), 1211 (C–O cyclic ether), 1194 (C–O ester). $^1$**H NMR** (300 MHz, DMSO, TMS): $\delta$ (ppm) = 1.22 (s, 1H, –OCH$_3$), 1.09 (s, 3H, –CCH$_3$), 1.13 (s, 3H, –CCH$_3$), 1.48,1.59 (sextet, 2H, 7.8, 7.5 Hz, –CH$_2$CH$_2$C=O), 3.08 (t, 2H, 7.5 Hz, –NCH$_2$CH$_2$), 5.92 (d, 1H, 10.5 Hz, –CCHCH), 6.69 (d,1H, 7.5 Hz, ArH), 6.81 (t, 1H, 7.8 Hz, ArH), 6.86 (d, 1H, 9 Hz, ArH), 7.08 (t, 1H, 7.5 Hz, ArH), 7.22 (d, 1H, 10.5 Hz, –CHCHC), 7.42 (d, 1H, 7.2 Hz, ArH), 7.99, 8.01, (dd, 1H, 2.7, 9 Hz, ArH), 8.22 (d, 1H, 2.7 Hz, ArH). $^{13}$**C NMR** (75 MHz, DMSO): $\delta$ (ppm) = 14.4, 19.9, 26.4, 44.6, 52.7, 61.0, 105.7, 107.2, 116.1, 119.1, 119.9, 121.2, 122.1, 123.3, 126.2, 127.9, 129.1, 135.6, 141.0, 146.2, 159.3, 170.1.

### 2.4.7. Synthesis of 1′,3′-dihydro-3′,3′-dimethyl-1′-ethyl-6-nitro-spiro(2H-1-benzopyran-2,2′-(2H)-indole) (SP-7)

The compound (SP-7) was synthesized by the same general procedure as given before, using 1-ethyl-3,3-dimethyl-2-methyleneindoline (TH-1) (0.424 g, 2.26 mmol), EtOH (15 ml) and 2-hydroxy-5-nitrobenzaldehyde (0.300 g, 1.79 mmol).

Brown solid was obtained with a yield of 79%. **IR** ($\bar{\upsilon}_{max}$, cm$^{-1}$) 1651 (C=C aliphatic), 1610 (C=C aromatic), 1332 (C–N), 1212 (C–O cyclic ether). $^1$**H NMR** (300 MHz, DMSO): $\delta$ (ppm) = 1.08 (t, 3H, 7.5 Hz, –NCH$_2$CH$_3$), 1.08 (s, 3H, –CCH$_3$), 1.20 (s, 3H, –CCH$_3$), 3.16, (q, 2H, 7.2 Hz, –NCH$_2$CH$_3$), 6.00 (d, 1H, 10.5 Hz, –CCHCH), 6.60 (d, 1H, 9 Hz, ArH), 6.776 (t, 1H, 8.9 Hz, ArH), 6.86 (d, 1H, 9 Hz, ArH), 7.10 (t, 1H, 9 Hz, ArH), 7.11 (d, 1H, 8.7 Hz, ArH), 7.20 (d, 1H, 10.5 Hz, –CHCHC), 7.98, 8.01 (dd, 1H, 3,9 Hz, ArH), 8.22 (d, 1H, 3 Hz, ArH). $^{13}$**C NMR** (75 MHz, DMSO): $\delta$ (ppm) = 14.7, 19.9, 22.3, 44.9, 45.0, 105.9, 106.9, 115.8, 119.2, 119.34, 122.1, 122.3, 123.2, 123.3, 126.1, 128.1, 128.4, 140.4, 147.3, 160.0.

### 2.4.8. Synthesis of 1′,3′-dihydro-3′,3′-dimethyl-1′-propyl-6-nitro-spiro(2H-1-benzopyran-2,2′-(2H)-indole) (SP-8)

The compound (SP-8) was synthesized by the same general procedure as given before, using 3,3-dimethyl-2-methylene-1-propylindoline (TH-2) (0.398 g, 2.0 mmol), EtOH (15 ml) and 2-hydroxy-5-nitrobenzaldehyde (0.300 g, 1.79 mmol).

Blue bell solid was obtained with yield of 88%. **IR** ($\bar{\upsilon}_{max}$, cm$^{-1}$) 1649 (C=C aliphatic), 1608 (C=C aromatic), 1333 (C–N), 1209 (C–O cyclic ether). $^1$**H NMR** (300 MHz, DMSO): $\delta$ (ppm) = 0.84 (t, 3H, 7.2 Hz, –CH$_2$CH$_3$) 1.10 (s, 3H, –CCH$_3$), 1.20 (s, 3H, –CCH$_3$),1.59, (m, 2H, 7.5 Hz, –CH$_2$CH$_2$CH$_3$), 3.08 (t, 2H, 7.5 Hz, –NCH$_2$CH$_2$), 6.00 (d, 1H, 10.5 Hz, –CCHCH), 6.60 (d, 1H, 8.1 Hz, ArH), 6.86 (d, 1H, 9 Hz, ArH), 6.77 (t, 1H, 7.2 Hz, ArH), 7.10 (t, 1H, 6 Hz, ArH), 7.11 (d, 1H, 8.7 Hz, ArH), 7.20 (d, 1H, 10.5 Hz, –CHCHC), 7.98, 8.01 (dd, 1H, 3, 9 Hz, ArH), 8.221 (d, 1H, 3 Hz, ArH). $^{13}$**C NMR** (75 MHz, DMSO): $\delta$ (ppm) = 11.9, 19.9, 22.1, 26.3, 45.0, 52.2, 106.8, 106.9, 115.8, 119.1, 119.3, 122.1, 122.3, 123.3, 123.3, 126.1, 128.0, 128.5, 140.8, 147.2, 159.6.

### 2.4.9. Synthesis of 1′,3′-dihydro-3′,3′-dimethyl-1′-butyl-6-nitro-spiro(2H-1-benzopyran-2,2′-(2H)-indole) (SP-9)

The compound (SP-9) was synthesized by the same general procedure as given before, using 1-butyl-3,3-dimethyl-2-methyleneindoline (TH-3) (0.663 g, 3.0 mmol), EtOH (15 ml), and 2-hydroxy-5-nitrobenzaldehyde (0.300 g, 1.79 mmol).

Coral pink solid was obtained with yield of 85%. **IR** ($\bar{\upsilon}_{max}$, cm$^{-1}$) 1651 (C=C aliphatic), 1609 (C=C aromatic), 1333 (C–N), 1207 (C–O cyclic ether). $^1$**H NMR** (300 MHz, DMSO): $\delta$ (ppm) = 0.83 (t, 3H,

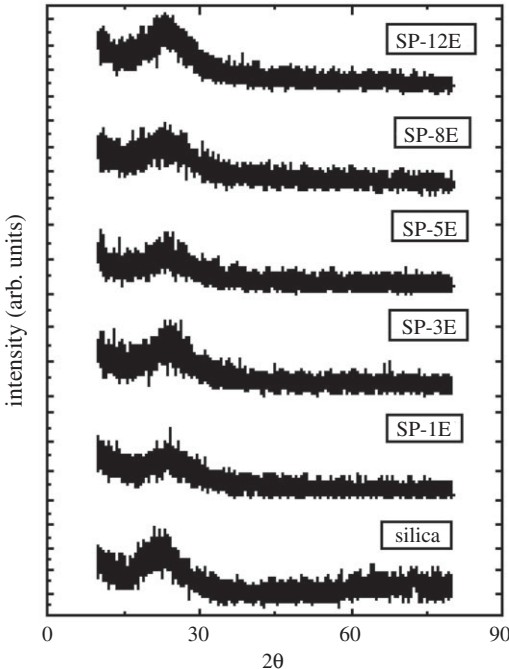

**Figure 3.** XRD patterns of hollow silica mesospheres and dye encapsulated mesospheres.

7.5 Hz, –CH$_2$CH$_3$), 1.20 (s, 3H, –CCH$_3$), 1.24 (s, 3H, –CCH$_3$), 1.28 (sextet, 2H, 7.5 Hz, –CH$_2$CH$_2$CH$_3$), 1.52 (quintet, 2H, 7.5 Hz, –CH$_2$CH$_2$CH$_2$), 3.21 (t, 2H, 7.5 Hz, –NCH$_2$CH$_2$), 6.00 (d, 1H, 10.5 Hz, –CCHCH), 6.59 (d, 1H, 9 Hz, ArH), 6.77 (t, 1H, 8.9 Hz, ArH), 6.86 (d, 1H, 9 Hz, ArH), 7.10 (t, 1H,9 Hz, ArH), 7.21 (d, 1H, 10.5 Hz, –CHCHC), 7.42 (d, 1H, 9 Hz, ArH), 7.98, 8.00 (dd, 1H, 2.7, 9 Hz, ArH), 8.22 (d, 1H, 2.7 Hz, ArH). **$^{13}$C NMR** (75 MHz, DMSO): $\delta$ (ppm) = 14.0, 18.0, 19.0, 19.9, 29.4, 48.9, 51.9, 105.0, 106.8, 117.2, 118.9, 119.9, 121.2, 121.9, 123.3, 126.1, 127.9, 129.1, 136.0, 141.1, 146.2, 159.3.

## 2.4.10. Synthesis of 1′,3′-dihydro-1′-(1-methylpropyl)-3′,3′-dimethyl-6-nitro-spiro(2H-1-benzopyran-2,2′-(2H)-indole) (SP-10)

The compound (SP-10) was synthesized by the same general procedure as given before, using 1-sec-butyl-3,3-dimethyl-2-methyleneindoline (TH-4) (0.540 g, 2.5 mmol), EtOH (15 ml), and 2-hydroxy-5-nitrobenzaldehyde (0.300 g, 1.79 mmol).

Citrine solid was obtained with yield of 80%. **IR** ($\bar{\upsilon}_{max}$, cm$^{-1}$) 1641 (C=C aliphatic), 1610 (C=C aromatic), 1334 (C–N), 1211 (C–O cyclic ether). **$^1$H NMR** (300 MHz, DMSO): $\delta$ (ppm) = 0.85 (t, 3H, 7.5 Hz, –CH$_2$CH$_3$), 1.09 (s, 3H, –CCH$_3$), 1.13 (s, 3H, –CCH$_3$), 1.25 (d, 3H, 7.5 Hz, –CHCH$_3$), 1.53 (quintet, 2H, 7.5 Hz, –CHCH$_2$CH$_3$), 2.79 (sextet, H, 7.8 Hz, –NCH(CH$_2$)(CH$_3$)), 6.00 (d, 1H, 10.5 Hz, –CCHCH), 6.57 (d, 1H, 9 Hz, ArH), 6.81 (t, 1H, 8.8 Hz, ArH), 6.87 (d, 1H, 9 Hz, ArH), 7.08 (t, 1H, 8.9 Hz, ArH), 7.22 (d, 1H, 10.5 Hz, –CHCHC), 7.42 (d, 1H, 9 Hz, ArH), 7.98, 8.01 (dd, 1H, 3, 9 Hz, ArH), 8.23 (d, 1H, 3 Hz, ArH). **$^{13}$C NMR** (75 MHz, DMSO): $\delta$ (ppm) = 14.0, 17.4, 19.9, 21.0, 30.4, 49.9, 52.7, 106.0, 106.8, 117.0, 119.1, 119.9, 122.0, 121.2, 123.3, 126.1, 127.9, 129.1, 135.4, 141.0, 146.2, 159.3.

## 2.4.11. Synthesis of 1′,3′-dihydro-1′-(1-methylacetate)-3′,3′-dimethyl-6-nitro-spiro(2H-1-benzopyran-2,2′-(2H)-indole) (SP-11)

The compound (SP-11) was synthesized by the same general procedure as given before, using methyl-2-(3,3-dimethyl-2-methyleneindolin-1-yl)acetate (TH-5) (0.582 g, 2.5 mmol), EtOH (15 ml), and 2-hydroxy-5-nitrobenzaldehyde (0.412 g, 2.4 mmol).

Pink solid was obtained with yield of 70%. **IR** ($\bar{\upsilon}_{max}$, cm$^{-1}$) 1657 (C=C aliphatic),1742 (C=C ester), 1612 (C=C aromatic), 1336 (C–N), 1210 (C–O cyclic ether), 1193 (C–O ester). **$^1$H NMR** (300 MHz, DMSO, TMS): $\delta$ (ppm) = 1.08 (t, 3H, 7.5 Hz, –OCH$_3$), 1.19 (s, 3H, –CCH$_3$), 1.30 (s, 3H, –CCH$_3$), 2.75 (s, 2H, –NCH$_2$C=O), 6.00 (d, 1H, 10.5 Hz, –CCHCH), 6.60 (d, 1H, 9 Hz, ArH), 6.77 (t, 1H, 8.9 Hz, ArH), 6.86 (d, 1H, 9 Hz, ArH), 7.10 (t, 1H, 9 Hz, ArH), 7.11 (d, 1H, 8.7 Hz, ArH), 7.20 (d, 1H, 10.5 Hz,

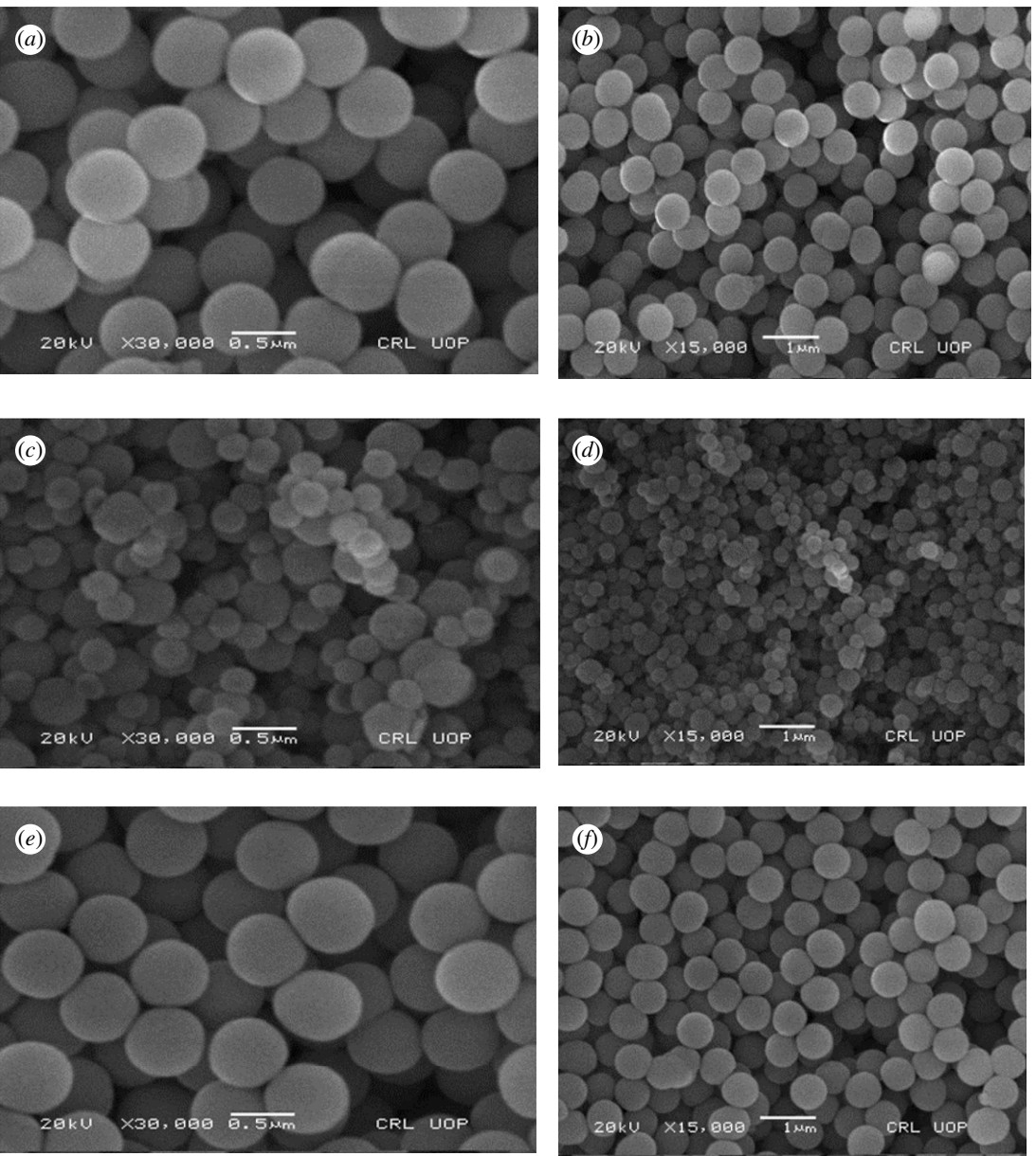

**Figure 4.** SEM images of various encapsulated spiropyrans mesospheres. (*a*) SEM of SP-3E at a magnification of 30 000. (*b*) SEM of SP-3E at a magnification of 15 000. (*c*) SEM of SP-8E at a magnification of 30 000. (*d*) SEM of SP-8E at a magnification of 15 000. (*e*) SEM of SP-12E at a magnification of 30 000. (*f*) SEM of SP-12E at a magnification of 15 000.

–CH*CHC*), 7.98, 8.01 (dd, 1H, 3, 9 Hz, Ar*H*), 8.22 (d, 1H, 3 Hz, Ar*H*). **13C NMR** (75 MHz, DMSO, TMS): $\delta$(ppm) = 19.9, 22.1, 44.9, 52.0, 54.0, 106.8, 106.9, 115.8, 119.2, 119.3, 122.1, 122.3, 123.2, 123.3, 126.1, 128.1, 128.4, 141.0, 147.2, 159.7, 172.0.

## 2.4.12. Synthesis of 1′,3′-dihydro-1′-(1-ethylacetate)-3′,3′-dimethyl-6-nitro-spiro(2H-1-benzopyran-2,2′-(2H)-indole) (SP-12)

The compound (SP-12) was synthesized by the same general procedure as given before, using methyl-3-(3,3-dimethyl-2-methyleneindolin-1-yl)propanoate (TH-6) (0.747 g, 3.0 mmol), EtOH (15 ml), and 2-hydroxy-5-nitrobenzaldehyde (0.300 g, 1.79 mmol).

Brown solid was obtained with yield of 81%. **IR** ($\bar{\upsilon}_{max}$, cm$^{-1}$) 1642 (C=C aliphatic), 1606 (C=C aromatic), 1336 (C–N), 1213 (C–O cyclic ether). **1H NMR** (300 MHz, DMSO): $\delta$ (ppm) = 1.22 (s, 1H, –OC*H₃*), 1.09 (s, 3H, –CC*H₃*), 1.13 (s, 3H, –CC*H₃*), 4.00 (t, 2H, 6.9 Hz, –CH₂C*H₂*C=O), 4.03 (t, 2H, 6.9 Hz, –NC*H₂*CH₂), 5.92 (d, 1H, 10.5 Hz, –CC*H*CH), 6.57 (t, 1H, 7.8 Hz, Ar*H*), 6.81 (d, 1H, 8.8 Hz, Ar*H*), 6.87 (d, 1H, 9 Hz, Ar*H*), 7.10 (d, 1H, 7.5 Hz, Ar*H*), 7.13 (t, 1H, 7.5 Hz, Ar*H*), 7.22 (d, 1H,

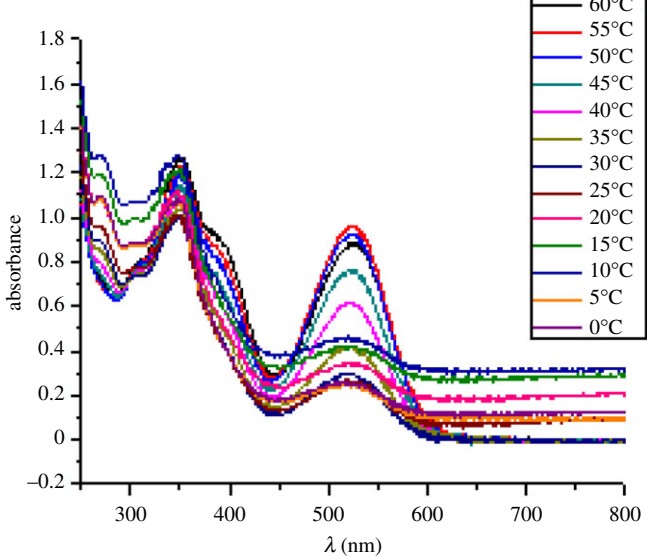

**Figure 5.** Temperature-dependent change in absorption spectra of SP-8 (e50 μM) measured in water-MeOH (1 : 1 v/v).

10.5 Hz, –CH*CHC*), 7.99, 8.02, (dd, 1H, 2.7, 9 Hz, Ar*H*), 8.22 (d, 1H, 2.7 Hz, Ar*H*). $^{13}$**C NMR** (75 MHz, DMSO): $\delta$ (ppm) = 14.4, 19.9, 26.47, 44.6, 52.7, 61.0, 106.0, 107.2, 117.0, 119.1, 119.9, 121.2, 122.0, 123.3, 126.1, 127.9, 129.1, 135.4, 141.0, 146.2, 159.3, 170.1.

## 2.5. General procedure for the encapsulation of synthesized spiropyrans

The synthesized spiropyrans (50 mmol) were dissolved in ethanol (A) (5 ml). This solution (A) (4.6 ml) was added to the ethanol-water mixture (8 : 2 ratio). This reaction mixture was then stirred for 20 min followed by the addition of TEOS (5 ml) and ammonium hydroxide (1 ml). The reaction was stirred for 48 h, and the product was obtained by filtration, washed with ethanol and dried in a vacuum.

# 3. Results and discussion

In this work, the synthesis of novel silica-encapsulated spiropyrans is reported. Preparation of silica NPs as well-defined structures requires precise planning and depends upon a number of factors such as selection of proper precursor, molar ratio, temperature, reaction time, stirring rate and rate of addition of precursor to the reaction. All these factors contribute to the final shape and morphology of the silica NP [37]. The type of bonding between the spiropyran and silica is mainly electrostatic interaction, hydrogen bonding and other non-covalent interactions [38]. The target spiropyrans were synthesized through one of the standard methods by the condensation of methylene bases (or their precursors) with *o*-hydroxy aromatic [39]. The structures of synthesized compounds were characterized by spectroscopic techniques like IR, $^1$H NMR and $^{13}$C NMR spectroscopic techniques. All compounds gave satisfactory elemental analysis, and the observed percentage of carbon and hydrogen atoms was found in good agreement with the calculated values.

The IR absorption spectrum of the different spiropyrans showed characteristic bands in the range of 1620–1680, 1310–1360, 1200–1300, 1735–1750, 1150–1250 cm$^{-1}$, attributed to C=C (aliphatic), C–N, C–O (ether), C=O (ester) and C–O (ether), respectively [40]. The IR spectroscopic analyses of the encapsulated spiropyrans were also carried out. The characteristic peak for Si-O-Si and Si-O-H bending vibrations appeared in 900–1460 cm$^{-1}$ range of all the IR spectra, whereas the O–H stretching vibration was observed at an average of 3200 cm$^{-1}$ [41].

The synthesized mesospheres were also characterized via X-ray diffraction (XRD) technique. The XRD spectra of some of the dye encapsulated mesospheres are shown in figure 3. No characteristic Bragg diffraction peaks at 2θ below 10° are observed, whereas a broad peak with 2θ centred at 21° is found. This peak indicates the disordered nature of the mesosphere [35]. The XRD of simple silica mesospheres without the dye is compared with those having the dye. The observed pattern is almost similar in both cases, which shows that the dye is successfully encapsulated in the silica mesospheres and is not attached to the outer bounderies of the silica shell [42].

**Table 2.** Temperature-dependent transformation of target compounds (SP1–SP12).

| sample code | thermochromic transformation |
|---|---|
| SP1 |  |
| SP2 |  |
| SP3 |  |
| SP4 |  |

(*Continued.*)

**Table 2.** (*Continued.*)

| sample code | thermochromic transformation |
|---|---|
| SP5 | 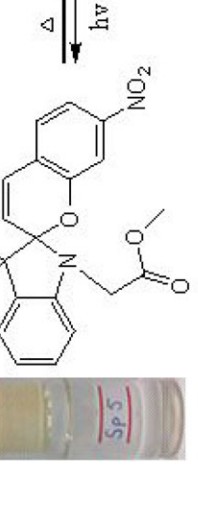 |
| SP6 | 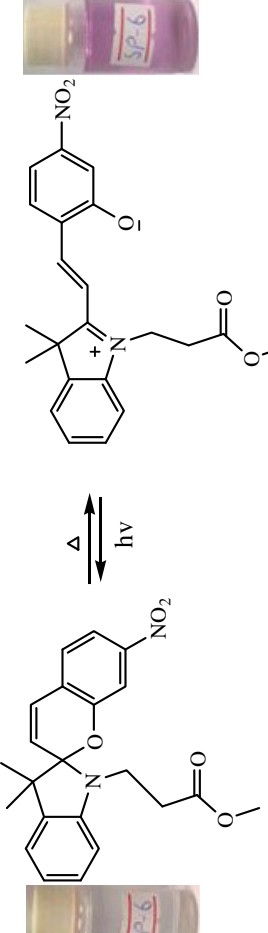 |
| SP7 | 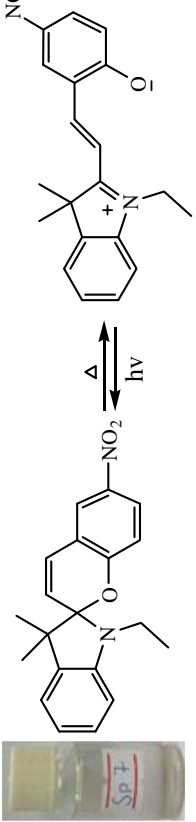 |

*(Continued.)*

**Table 2.** *(Continued.)*

| sample code | thermochromic transformation |
|---|---|
| SP8 |  |
| SP9 |  |
| SP10 |  |

**Table 2.** (*Continued.*)

| sample code | thermochromic transformation |
| --- | --- |
| SP11 |  |
| SP12 |  |

SEM micrographs of the encapsulated mesosphere (figure 4) represent that they are spherical and homogeneous with their particle size in the range of 450–500 nm. All the particles are non-porous as no surfactant was used in the synthesis of mesospheres [43]. During synthesis, the reaction mixture was stirred very slowly, so that the mesoparticles are obtained. The reason being the factor that spiropyrans required some space so that they can easily convert thermally from SP to MC form.

The temperature-dependent change in absorption spectra of SP-8 (50 μM) measured in water-MeOH (1 : 1 v/v) is shown (figure 5). Spiropyran shows two different interconvertible absorption peaks in the UV-visible region, each corresponding to the SP and MC forms. At a temperature of 0°C, a peak at 350 nm is observed which corresponds to the SP form, whereas a peak of comparatively smaller intensity is observed in the range of 450–600 nm which corresponds to the MC form. The appearance of this peak is due to the photochromic behaviour of the synthesized spiropyrans. As the temperature was increased after regular intervals, the absorption intensity of the MC form also increased, and this trend conforms to the thermochromic behaviour of the as-synthesized spiropyrans. It is observed that the increase in absorption is very slow up to 25°C. However, the increase in absorption spectra becomes more prominent with a further rise in temperature. The MC form is stabilized in the solution due to the hydrogen bonding. The colour of the solution changes from very light pink to dark pink with a temperature rise [44].

The temperature-dependent transformation of all the synthesized spiropyrans is shown in table 2.

# 4. Conclusion

Indoles were substituted with different alkyl and ester moieties by a simple nucleophilic reaction. The substituted indoles were then converted to 12 novel spiropyrans by the condensation of methylene base with nitro-substituted *o*-hydroxy aromatic aldehydes. All the synthesized spiropyrans were characterized by physical data, $^1$H NMR, $^{13}$C NMR and Fourier transform infrared (FTIR) spectroscopy. All the spiropyrans were solids with different colours and show characteristic peaks in FTIR data. The structural novelty of the prepared spiropyrans was confirmed by the $^1$H NMR and $^{13}$C NMR. These spiropyrans were encapsulated within the nanospheres of the slice by using tetraethylorthosilicate as a silica precursor. XRD showed that the spiropyrans were fully incorporated within the silica mesospheres. Encapsulation of thermochromic dyes was confirmed by the SEM and FTIR techniques. SEM shows the non-porous mesospheres within the size range of 450–500 nm. The thermochromic properties of these spiropyrans were confirmed by the UV-Vis spectroscopy, in the temperature range of 0–50°C, and discontinuous change in the absorbance spectra of spiropyrans was observed with temperature increase. Overall, we have effectively developed a metal oxide encapsulated spiropyrans as efficient temperature indicator in the temperature range of 0–5°C.

Data accessibility. Electronic supplementary material can be found via Zenodo: https://zenodo.org/record/6284895.
Authors' contributions. A.I.: conceptualization, methodology, writing—original draft; G.I.: data curation, methodology; M.N.U.: software, validation; H.u.R.: writing—review and editing; S.W.K.: data curation, formal analysis, investigation.
All authors gave final approval for publication and agreed to be held accountable for the work performed therein.
Competing interests. The authors declare that they have no conflict of interest.
Funding. This research was funded by Nanoscience and Technology Department, National Center for Physics, Islamabad, Pakistan.
Acknowledgements. The authors are thankful to Dr Samina Nazir (Assistant Prof., King Faisal University, Saudi Arabia) for her support to carry out this research. The authors also express their thanks to the anonymous reviewers who provided valuable comments and suggestions to improve the quality of their manuscript.

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
