## [Peer Review File · Royal Society Open Science]

Review History

RSOS-211385.R0 (Original submission)

Review form: Reviewer 1

Is the manuscript scientifically sound in its present form?

Yes

Are the interpretations and conclusions justified by the results?

Yes

Is the language acceptable?

Yes

Do you have any ethical concerns with this paper?

No

Have you any concerns about statistical analyses in this paper?

No

Recommendation?

Accept with minor revision (please list in comments)

Comments to the Author(s)

Manuscript ID: RA-ART-08-2021-006341 -----

Reviewer's Comment 1: The manuscript has deficiency citation to similar works published before like: Encapsulation of photochromic compounds possessing positive and negative photochromism. Materials letters Vol 303, 15 November 2021, 130558, Spiropyran based drug delivery systems, review article, front. chem. 29 July 2021.

The evolution of spiropyran: fundamentals and progress of an extraordinarily versatile photochrome. Chemical society reviews, Issue 12, 2019.

Reviewer's Comment 2: Factors effecting the size of mesosphers should be discussed in the manuscript.

Reviewer's Comment 3: The Type of bonding between the spropyran and silica should be elaborated.

Reviewer's Comment 4: Elaborate the formation of silica mesospheres.

Reviewer's Comment 5: The applications of the synthesized silica encapsulated spiropyran with emphasis to the thermochromic behavior should be Discussed.

Review form: Reviewer 2

Is the manuscript scientifically sound in its present form?

Yes

Are the interpretations and conclusions justified by the results?

Yes

Is the language acceptable?

Yes

Do you have any ethical concerns with this paper?

No

Have you any concerns about statistical analyses in this paper?

No

Recommendation?

Major revision is needed (please make suggestions in comments)

Comments to the Author(s)

Following points should be encountered prior to publication:

FTIR and elemental analysis and interpretation should complete the work.

Particle size distribution of the silica particles should be presented.

Applicability of the particles is very much important. What is produced could only be meaningful when applied and tested in an application field.

Other minor points are;

Abbreviation should be placed first before use.

Spaces should be used conveniently in the text.

Number of digits should be same after dot along with the text.

Figure 4 should be reconfigured to be clear enough.
Figure 5 is not clear.

Decision letter (RSOS-211385.R0)

Dear Dr ur Rashida:

Title: Synthesis of Novel Silica Encapsulated Spiropyran-based ThermoChromic Materials
Manuscript ID: RSOS-211385

The editor assigned to your manuscript has now received comments from reviewers. We would like you to revise your paper in accordance with the referee and Subject Editor suggestions which can be found below (not including confidential reports to the Editor). Please note this decision does not guarantee eventual acceptance.

Please submit your revised paper before 19-Nov-2021. Please note that the revision deadline will expire at 00.00am on this date. If we do not hear from you within this time then it will be assumed that the paper has been withdrawn. In exceptional circumstances, extensions may be possible if agreed with the Editorial Office in advance. We do not allow multiple rounds of revision so we urge you to make every effort to fully address all of the comments at this stage. If deemed necessary by the Editors, your manuscript will be sent back to one or more of the original reviewers for assessment. If the original reviewers are not available we may invite new reviewers.

Yours sincerely,
Dr Ellis Wilde
Publishing Editor, Journals

On behalf of the Subject Editor Professor Anthony Stace and the Associate Editor Professor Chaohua Cui.

RSC Associate Editor
Comments to the Author:
(There are no comments.)

RSC Subject Editor
Comments to the Author:
(There are no comments.)

Reviewers' Comments to Author:

Reviewer: 1

Comments to the Author(s)

Manuscript ID: RA-ART-08-2021-006341 -----

Reviewer's Comment 1: The manuscript has deficiency citation to similar works published before like: Encapsulation of photochromic compounds possessing positive and negative photochromism. Materials letters Vol 303, 15 November 2021, 130558, Spiropyran based drug delivery systems, review article, front. chem. 29 July 2021.

The evolution of spiropyran: fundamentals and progress of an extraordinarily versatile photochrome. Chemical society reviews, Issue 12, 2019.

Reviewer's Comment 2: Factors effecting the size of mesosphers should be discussed in the manuscript.

Reviewer's Comment 3: The Type of bonding between the spropyran and silica should be elaborated.

Reviewer's Comment 4: Elaborate the formation of silica mesospheres.

Reviewer's Comment 5: The applications of the synthesized silica encapsulated spiropyran with emphasis to the thermochromic behavior should be Discussed.

Reviewer: 2

Comments to the Author(s)

Following points should be encountered prior to publication:

FTIR and elemental analysis and interpretation should complete the work.

Particle size distribution of the silica particles should be presented.

Applicability of the particles is very much important. What is produced could only be meaningful when applied and tested in an application field.

Other minor points are;

Abbreviation should be placed first before use.

Spaces should be used conveniently in the text.

Number of digits should be same after dot along with the text.

Figure 4 should be reconfigured to be clear enough.

Figure 5 is not clear.

Author's Response to Decision Letter for (RSOS-211385.R0)

See Appendix A.

RSOS-211385.R1 (Revision)

Review form: Reviewer 1

Is the manuscript scientifically sound in its present form?

Yes

Are the interpretations and conclusions justified by the results?

Yes

Is the language acceptable?

Yes

Do you have any ethical concerns with this paper?

No

Have you any concerns about statistical analyses in this paper?

No

Recommendation?

Accept as is

Comments to the Author(s)

Accept

Review form: Reviewer 2

Is the manuscript scientifically sound in its present form?

Yes

Are the interpretations and conclusions justified by the results?

Yes

Is the language acceptable?

Yes

Do you have any ethical concerns with this paper?

No

Have you any concerns about statistical analyses in this paper?

No

Recommendation?

Major revision is needed (please make suggestions in comments)

Comments to the Author(s)

Following points should be considered prior to publications:

Intro of the manuscript should be improved with the studies about microencapsulated thermochromic systems.

State of the art microencapsulated thermochromic dyes should be combined in the text and literature library. There are PMMA based particles and the authors should compare the present study to those also. Conclusion part should be enriched with the findings remarkable to literature.

Thermal property of different type of spiropyranes and the microencapsulated species should be investigated through DSC and TGA investigations.

Review form: Reviewer 3 (Kamal I Aly)

Is the manuscript scientifically sound in its present form?

Yes

Are the interpretations and conclusions justified by the results?

Yes

Is the language acceptable?

Yes

Do you have any ethical concerns with this paper?

No

Have you any concerns about statistical analyses in this paper?

No

Recommendation?

Accept as is

Comments to the Author(s)

Dear Professor,

Thank you for your remind me for sending my comment.

Manuscript title: Synthesis of Novel Silica Encapsulated Spiropyran-based Thermochromic Materials

Manuscript ID: RSOS-211385

I think that the authors were answered on all the comments from all the reviewers, and this is sufficient to published this manuscript.

I recommend to accept this paper in the Royal Society open Science.

Best Regard

Kamal I Aly

Decision letter (RSOS-211385.R1)

Dear Dr ur Rashida:

Title: Synthesis of Novel Silica Encapsulated Spiropyran-based Thermochromic Materials
Manuscript ID: RSOS-211385.R1

Thank you for submitting the above manuscript to Royal Society Open Science. On behalf of the Editors and the Royal Society of Chemistry, I am pleased to inform you that your manuscript will be accepted for publication in Royal Society Open Science subject to minor revision in accordance with the referee suggestions. Please find the reviewers' comments at the end of this email.

The reviewers and handling editors have recommended publication, but also suggest some minor revisions to your manuscript. Therefore, I invite you to respond to the comments and revise your manuscript.

Please also include the following statements alongside the other end statements. As we cannot publish your manuscript without these end statements included, if you feel that a given heading is not relevant to your paper, please nevertheless include the heading and explicitly state that it is not relevant to your work. We have included a screenshot example of the end statements for reference.

- Ethics statement

Please clarify whether you received ethical approval from a local ethics committee to carry out your study. If so please include details of this, including the name of the committee that gave consent in a Research Ethics section after your main text. Please also clarify whether you received informed consent for the participants to participate in the study and state this in your Research Ethics section.

OR

Please clarify whether you obtained the necessary licences and approvals from your institutional animal ethics committee before conducting your research. Please provide details of these licences and approvals in an Animal Ethics section after your main text.

OR

Please clarify whether you obtained the appropriate permissions and licences to conduct the fieldwork detailed in your study. Please provide details of these in your methods section.

- Data accessibility

It is a condition of publication that you make available the data and research materials supporting the results in the article. Datasets should be deposited in an appropriate publicly available repository and details of the associated accession number, link or DOI to the datasets must be included in the Data Accessibility section of the article (<https://royalsocietypublishing.org/rsos/for-authors#question17>). Reference(s) to datasets should also be included in the reference list of the article with DOIs (where available).

Please include a Data Availability section after your main text stating where supporting data are available from, or where they will be made available should your article be accepted for publication.

If you wish to submit your supporting data or code to Dryad (<http://datadryad.org/>), or modify your current submission to dryad, please use the following link:
<http://datadryad.org/submit?journalID=RSOS&manu=RSOS-211385.R1>

- **Competing interests**

Please include a Competing Interests section after your main text declaring any financial or non-financial competing interests. If you have no competing interests please state 'I/we have no competing interests.'

- **Authors' contributions**

Please include an Authors' Contributions section at the end of your main text detailing the contribution of each author. All authors should have read and approved the manuscript before submission and this should be stated in the Authors' Contributions section.

The list of Authors should meet all of the following criteria; 1) substantial contributions to conception and design, or acquisition of data, or analysis and interpretation of data; 2) drafting the article or revising it critically for important intellectual content; and 3) final approval of the version to be published.

- **Acknowledgements**

- **Funding statement**

Please include a funding section after your main text which lists the source of funding for each author.

Because the schedule for publication is very tight, it is a condition of publication that you submit the revised version of your manuscript before 23-Jan-2022. Please note that the revision deadline will expire at 00.00am on this date. If you do not think you will be able to meet this date please let me know immediately.

Kind regards,
Dr Ellis Wilde
Publishing Editor, Journals

On behalf of the Subject Editor Professor Anthony Stace and the Associate Editor Professor Chaohua Cui.

RSC Associate Editor
Comments to the Author:
(There are no comments.)

RSC Subject Editor
Comments to the Author:
(There are no comments.)

Important Note: THE COMMENTS OF REVIEWER 1 HAVE BEEN REMOVED DUE TO A POTENTIAL CONFLICT OF INTEREST.

Reviewer: 2

Comments to the Author(s)

Following points should be considered prior to publications:

Introduction of the manuscript should be improved with the studies about microencapsulated thermochromic systems.

State of the art microencapsulated thermochromic dyes should be combined in the text and literature library. There are PMMA based particles and the authors should compare the present study to those also. Conclusion part should be enriched with the findings remarkable to literature.

Thermal property of different type of spiropyranes and the microencapsulated species should be investigated through DSC and TGA investigations.

Reviewer: 3

Comments to the Author(s)

Dear Professor,

Thank you for your remind me for sending my comment.

Manuscript title: Synthesis of Novel Silica Encapsulated Spiropyran-based Thermochromic Materials

Manuscript ID: RSOS-211385

I think that the authors were answered on all the comments from all the reviewers, and this is a sufficient to published this manuscript.

I recommend to accept this paper in the Royal Society open Science.

Best Regard

Kamal I Aly

Author's Response to Decision Letter for (RSOS-211385.R1)

See Appendix B.

Decision letter (RSOS-211385.R2)

Dear Dr ur Rashida:

Title: Synthesis of Novel Silica Encapsulated Spiropyran-based Thermochromic Materials

Manuscript ID: RSOS-211385.R2

It is a pleasure to accept your manuscript in its current form for publication in Royal Society Open Science. The chemistry content of Royal Society Open Science is published in collaboration with the Royal Society of Chemistry.

Yours sincerely,
Dr Ellis Wilde
Publishing Editor, Journals

On behalf of the Subject Editor Professor Anthony Stace and the Associate Editor Professor Chaohua Cui.

RSC Associate Editor
Comments to the Author:
(There are no comments.)

Reviewer(s)' Comments to Author:

Appendix A

Response Letter to the Reviewers Comments

Manuscript title: Synthesis of Novel Silica Encapsulated Spiropyran-based Thermochromic Materials

Manuscript ID: RSOS-211385

We highly appreciate the comments on our manuscript. These comments were extremely helpful to improve the quality of manuscript. We addressed the comment from reviewers as following.

Note: Reviewers Comments are in bold font followed by the author's response in regular fonts. In the main manuscript file the response to the reviewers, points are highlighted yellow.

Response to the Reviewer 1:

Reviewer's Comment 1: The manuscript has deficiency citation to similar works published before like:

- (A) Encapsulation of photochromic compounds possessing positive and negative photochromism. Materials letters Vol 303, 15 November 2021, 130558.**
- (B) Spiropyran based drug delivery systems, review article, front. chem. 29 July 2021.**
- (C) The evolution of spiropyran: fundamentals and progress of an extraordinarily versatile photochrome. Chemical society reviews, Issue 12, 2019.**

Author's Response:

The above mentions references are added to the manuscript as

- (A) Reference no 16 Pg 2, line 12-14 under the heading of introduction as "All these properties have wide range of applications but photochromism and thermochromism of spiropyran have attracted wide range of applications such as in cosmetic, fabrics etc."
- (B) Reference no 17 Pg 2, line 14-16 under the heading of introduction as "Spiropyran are considered as a good candidate for application in drug delivery systems in which they are used incorporation with different polymeric micelles, polymersomes and polymeric nanoparticles etc."

(C) Reference 15 no Pg 2, line 10-12 under the heading of introduction, as “Spiropyrans are one of the contender of the most versatile compound reported as it shows diverse properties such as photochromism, fluorescence, mechanochromism, acidochromism etc.”

Reviewer’s Comment 2: Factors effecting the size of mesosphers should be discussed in the manuscript.

Author’s Response: The comments has been addressed, in reference 23, on Pg 15 line 3-6 under the heading of results and Discussion as “Preparation of silica nanoparticles as well defined structures require precise planning and depends upon a number of factors such as selection of proper precursor, molar ratio, temperature, reaction time, stirring rate and rate of addition of precursor to the reaction. All these factors contribute to the final shape and morphology of the silica nanoparticle.”

Reviewer’s Comment 3: The Type of bonding between the spropyran and silica should be elaborated.

Author’s Response: The comments has been addressed, in reference 24 on Pg 15, line 6-8 under the heading of results and Discussion as “The type of bonding between the spiropyran and silica is mainly electrostatic interaction, hydrogen bonding and other non-covalent interactions.”

Reviewer’s Comment 4: Elaborate the formation of silica mesospheres.

Author’s Response: The comments has been addressed on Pg 3 line 8-17 under the heading of introduction as “The stober method involve hydrolysis and condensation polymerization with TEOS under basic conditions as shown in the scheme below.

where R is an alkyl group C₂H₅. In the hydrolysis reaction (Eq. (A)), TEOS hydrolyzes to generate siloxane molecules and ethanol. Then the condensation polymerization occurs through

the siloxane molecules and TEOS molecules (Eq. (B)) or siloxane molecules themselves (Eq. (C))”

Reviewer’s Comment 5: The applications of the synthesized silica encapsulated spiropyran with emphasis to the thermochromic behavior should be Discussed.

Author’s Response: The comments as been addressed as reference 22 on Pg 3 line 21-24 under the heading of introduction as “The prepared silica encapsulated spiropyrans have wide range of applications as temperature monitoring sensors which are used in motors, circuit breakers, heat exchangers, transformers. These can be applied in health indicators and to report food quality. The general applications involve decorative use on cloths, utensils and paper etc”.

Response to the Reviewer 2:

Reviewer’s Comment 1: FTIR and elemental analysis and interpretation should complete the work

Author’s Response: the FTIR data of all the prepared samples from SP-1 to SP-12 is already discussed in their respective section under the heading of General Procedure for Synthesis of Spiropyrans. i.e.SP-1, Pg-7, Line-1-3, SP-2, Pg-7, Line-11-14, SP-3, Pg-7, Line-28-29, SP-4, Pg-8, Line-11-12, SP-5, Pg-8, Line -25-26, SP- 6, Pg-9, Line-8-10, SP-7, Pg-9, Line-22-23, SP-8, Pg-10, Line-6-7, SP-9, Pg-10, Line-20-21, SP-10, Pg-11, Line-5-6, SP-11, Pg-11, Line-18-20, SP-12, pg-12, Line -1-2.

Reviewer’s Comment 2: Particle size distribution of the silica particles should be presented.

Author’s Response:

The balance between monomer addition and nucleation determines the size distribution of particles and final particle size as the monomer was added slowly, same sized particles are obtained. The particle size measured from SEM studies is in the range of 450-600nm. All the particles are spherical and homogeneous with no pores as already discussed on pg 17 line 14 under the heading of results and discussions.

Reviewer's Comment 3: Applicability of the particles is very much important. What is produced could only be meaningful when applied and tested in an application field.

Author's Response: The synthesized materials show good thermochromic behavior with in the temperature range of 0-50°C. The experimental work for the application of these materials as temperature sensors is under progress and will be published soon.

Reviewer's Comment 4: Abbreviations should be placed first before use

Author's Response: All the abbreviations have been corrected.

Reviewer's Comment 5: Spaces should be used conveniently in the text.

Author's Response: All the spaces corrected.

Reviewer's Comment 6: Number of digits should be same after dot along with the text.

Author's Response: Number of digits corrected.

Reviewer's Comment 6: Figure 4 should be reconfigured to be clear enough.

Author's Response: Figure 4 corrected and reconfigured.

Reviewer's Comment 7: Figure 5 is not clear.

Author's Response: Figure 5 is magnified.

Appendix B

Response Letter to the Editor Comments

Manuscript title: Synthesis of Novel Silica Encapsulated Spiropyran-based Thermochromic Materials

Manuscript ID: RSOS-211385.R1-----

We highly appreciate the comments on our manuscript. These comments were extremely helpful to improve the quality of manuscript. We addressed the comment from reviewers as following.

Note: Reviewers Comments are in bold font followed by the author's response in regular fonts. In the main manuscript file the response to the reviewers, points are highlighted yellow.

Response to the Reviewer 2:

Reviewer's Comment 1: Intro of the manuscript should be improved with the studies about microencapsulated thermochromic systems. State of the art microencapsulated thermochromic dyes should be combined in the text and literature library.

Author's Response: The comments has been addressed, in reference 21-33, on Page 2 line 25 to page 3 line 21 and reference 34 on page 3 line-27-29 under the heading of introduction as “The main aim of encapsulation is to cloak any material (solid, liquid or gas) with in the coating material. The purposes of encapsulation can be different and depends on the specific applications.²¹ The main objective is always to protect the active material from external environment, which can affect the properties and applications of the main product (temperature, pH, etc.). It also helps to enhance the durability of the product by preventing the oxidation. Because of encapsulation, the core material properties are elevated.²² Different encapsulation techniques results in the formation of different capsules, and are classified by their size, i.e. microcapsules refers to the encapsulation where the size of these shells are less than 1 μm to 1000 μm in diameter and are of any type such as tubes, spheres, and the nanocapsules are synthesized at the nanoscale level.²³ Microencapsulation techniques consists of two main steps: emulsion formation/active ingredient suspension and shell formation. Emulsion preparation is

critical step as it determines the size distribution of the capsules, which are influenced by the operating conditions (the rate and time of agitation, viscosity, the mass ratio of different phases, etc.) Concentration of shell materials, pH, temperature and solubility effects the formation of the shell.²⁴ Different Encapsulations are used in pharmaceutical²⁵, cosmetic²⁶, and food industries.²⁷ Microencapsulation methods for encapsulating different thermochromic leuco dyes, and further incorporation of these microcapsules in smart coatings has been developed, which aids in making sustainable buildings that use minimum power for heating and cooling applications²⁸.

The microencapsulation of thermochromic materials by amorphous silica could protect it from UV light (290-320 nm).²⁹ A great selection of nanoparticles (NP's) such as Fe_3O_4 , SiO_2 , ZrO_2 , Al_2O_3 , and TiO_2 ³⁰⁻³² have been used for coating. Leuco dyes have been encapsulated with in silica to be use for energy saving in proposed building applications.³³

“In case of spiropyrans no such encapsulation with in the silica shells are reported, whereas research has been carried out on the encapsulation of a spiropyrans within a self-assembled cage.³⁵”

Reviewer’s Comment 2: There are PMMA based particles and the authors should compare the present study to those also.

Author’s Response: The comments has been addressed, in reference 34 on page 3 line 21-27 as “Poly(methylmethacrylate)/thermochromic system (PMMA/TS) microcapsules are synthesized by microencapsulation method and the thermochromic system was consisting of crystal violet lactone (CVL) as a leuco dye, bisphenolA (BPA) as a color developer, and 1-tetradecanol (TD) as a solvent. In case of PMMA/TS systems the ratio of PMMA to TS play a critical role, as the amount of PMMA increases the change in color become less visible due to the thickness of the shells. In comparison to the spiropyran these TS are multicomponent complex systems, which melts upon heating to at 45-50 °C.³⁴”

Reviewer's Comment 3: Conclusion part should be enriched with the findings remarkable to literature.

Author's Response: The comments has been addressed, on page 23 line 2-17 as “Indoles were substituted with different alkyl and ester moieties by a simple nucleophilic reaction. The substituted indoles were then converted to 12 novel Spiropyranes by the condensation of methylene base with nitro substituted *o*-hydroxy aromatic aldehydes. All the synthesized Spiropyranes were characterized by physical data, ¹H-NMR, ¹³C-NMR, and FTIR spectroscopy. All the spiropyranes were solids with different colors and show characteristic peaks in FTIR data. The structural novelty of the prepared spiropyranes were confirmed by the ¹H-NMR, and¹³C-NMR. These Spiropyranes were encapsulated within the nanospheres of the slice by using tetraethylorthosilicate as a silica precursor. XRD showed that the spiropyranes were fully incorporated within the silica mesospheres. Encapsulation of thermochromic dyes was confirmed by the SEM and FTIR techniques. SEM shows the non-porous mesospheres with in the size range of 450-500 nm. The thermochromic properties of these Spiropyranes were confirmed by the UV-Vis spectroscopy, in the temperature range of 0-50°C and discontinuous change in the absorbance spectra of Spiropyranes was observed with temperature increase. Overall, we have effectively developed a metal oxide encapsulated spiropyranes as efficient temperature indicator in the temperature range of 0-5°C”.

Reviewer's Comment 4: Thermal property of different type of spyropyranes and the microencapsulated species should be investigated through DSC and TGA investigations.

Author's Response: The main focus of present research work is to carry out the successful encapsulation of thermochromic spiropyranes. Further research work is in progress in which the synthesized microencapsulated spiropyranes will be incorporated in the matrix material to prepare temperature sensors strips/ films. All the mechanical and thermal properties will be discussed in that research work including TGA, DSC tensile strength etc.